# Complex Vascular Resections for Retroperitoneal Soft Tissue Sarcoma

**Nicolas A. Devaud** [1,*]**, Jean M. Butte** [1]**, Juan C. De la Maza** [2]**, Sebastian von Mühlenbrock Hugo** [2]
**and Kenneth Cardona** [3]

1  Sarcoma Surgery Unit, Instituto Oncologico Fundación Arturo Lopez Perez (Falp), Santiago 7500691, Chile
2  Vascular Surgery Unit, Instituto Oncologico Fundación Arturo Lopez Perez (Falp), Santiago 7500691, Chile
3  Division of Surgical Oncology, Department of Surgery, Winship Cancer Institute, Emory University School of Medicine, Atlanta, GA 30307, USA
*  Correspondence: nicolas.devaud@falp.org; Tel.: +56-961-290-512

**Abstract:** Retroperitoneal sarcomas (RPSs) are locally aggressive tumors that can compromise major vessels of the retroperitoneum including the inferior vena cava, aorta, or main tributary vessels. Vascular involvement can be secondary to the tumor's infiltrating growth pattern or primary vascular origin. Surgery is still the mainstay for curing this disease, and resection of RPSs may include major vascular resections to secure adequate oncologic results. Our improved knowledge in the tumor biology of RPSs, in conjunction with the growing surgical expertise in both sarcoma and vascular surgical techniques, has allowed for major vascular reconstructions within multi-visceral resections for RPSs with good perioperative results. This complex surgical approach may include the combined work of various surgical subspecialties.

**Keywords:** retroperitoneum; sarcoma; vascular reconstruction; oncovascular surgery

## 1. Introduction

Retroperitoneal soft tissue sarcomas (RPSs) are rare mesenchymal tumors that account for 15% of all soft tissue sarcomas (STSs). Among the different histological subtypes encountered in this location, retroperitoneal liposarcomas account for the highest incidence (60%) of all primary RPSs, followed by RP leiomyosarcomas (19%) [1]. There are tumor characteristics that define the primary growth and disease progression pattern within RPSs, such as histology type and tumor grade. High-grade tumors require a complex multimodal approach due to their recognized local and systemic patterns of failure [1–3]. Dedifferentiated liposarcomas (DD-LPSs) are the most common high-grade histologic subtype found in the retroperitoneum. Together with retroperitoneal leiomyosarcomas (RP-LMSs), they represent a group of high-grade tumors that may require extensive multi-visceral resections including complex vascular reconstruction such as the inferior vena cava (IVC), iliac vessels, or the aorta to attain an appropriate oncological resection. Other less frequent high-grade STSs found in the retroperitoneum such as undifferentiated pleomorphic sarcoma (UPS), solitary fibrous tumor (SFT), or malignant peripheral nerve sheath tumor (MPNST) may also require a vascular resection and reconstruction to secure clear margins.

The infiltrative growth pattern or vascular origin can impose a significant technical challenge on the surgical approach of these retroperitoneal tumors, requiring at times the combined expertise of Surgical oncologists together with vascular or transplant surgeons in highly specialized centers. This overview presents various aspects that should be taken in consideration in the treatment of RPSs that require complex vascular reconstruction.

## 2. Multimodal Approach: Biology vs. Surgical Risk Assessment

RPSs are commonly slow-growing and oligosymptomatic tumors that can present as a primary lesion, recurrent disease (after curative-intent surgery), or residual disease after an

incomplete resection. Less frequently, they can also present as a metastatic retroperitoneal lesion of a synchronous or previously treated extremity STS such as a myxoid/round cell liposarcoma [4,5].

Treatment of RPSs with major vascular involvement begins with a multidisciplinary assessment at a specialized sarcoma center. The assessment includes an evidence-based discussion amongst medical, radiation, and surgical oncologists, as well as sarcoma pathologists, toward a tailored patient- and histology-specific treatment plan [6,7]. To fulfill this purpose, a percutaneous pretreatment biopsy and updated staging imaging studies are paramount. As aforementioned, most frequent histology subtypes involving major vessels in the retroperitoneum are LMSs and DD-LPSs. It is important to note that well differentiated liposarcomas (WD-LPSs) can also involve major vascular structures of the retroperitoneum. However, their growth pattern is generally expansive and non-infiltrative; therefore, they generally do not require a vascular resection to secure R0 margins. WD-LPSs can generally be dissected from the vascular adventitia of the surrounded vessels without requiring their resection [8,9]. DD-LPSs, on the contrary, have an aggressive infiltrative growth pattern that may require resection and reconstruction of a surrounding major vessel of the retroperitoneum such as the IVC and/or the aorta.

The type of disease presentation is important to be considered. Primary high-grade RPSs have their best chance for cure after a well-planned index surgical resection. Although currently a topic of debate, high-grade primary RPSs such as DD-LPSs and LMSs may benefit from preoperative systemic chemotherapy [10]. When resectability appears to be borderline, neoadjuvant systemic therapy may induce various degrees of response that could secure the R0/R1 margin or identify those rapidly progressing tumors that present shortly with metastatic disease and, therefore, spare these patients from a futile high-risk operation. STRASS2 is a phase III randomized clinical trial (RCT) with the objective of elucidating a disease-specific survival (DSS) advantage after neoadjuvant combined anthracycline-based chemotherapy for high-grade RP-LPSs and RP-LMSs compared to upfront surgery.

The role of neoadjuvant radiation therapy in RPSs has also been under scrutiny as a way to improve local control after surgery. Local abdominal recurrent disease is common, presenting in over 50% of previously treated patients, with a significant number of them developing after 5 years of curative-intent resection [11]. Recurrent RPSs represent an important number of patients treated in sarcoma centers [12]; however, the advantage of neoadjuvant radiation to reduce local abdominal recurrence could not be proven for all histology subtypes after a 3 year follow-up in STRASS1[13]. This phase III RCT showed no benefit in local abdominal recurrence to surgery alone, although there was a tendency toward reduced recurrence among WD-LPSs after neoadjuvant radiation followed by surgery that could not reach statistical significance. Another recently published study comparing STRASS1 vs. off-trial patients who were not enrolled in STRASS1 at participating centers proved increased abdominal recurrence-free survival after neoadjuvant radiation followed by surgery among WD-LPSs and G1–G2 DD-LPSs (HR, 0.63; 95% CI, 0.40–0.97) [14]. In the context of WD-LPSs with major vascular involvement, this local control advantage may allow safely dissecting the tumor from the vascular adventitia with reduced risk of local failure.

In the setting of recurrent or residual disease after primary resection, surgery still provides a survival benefit in a select cohort of patients; thus, a close assessment of the biologic behavior and survival benefit should always be discussed when considering surgery within this setting, more so if complex vascular reconstruction is required. In cases of recurrent disease or residual tumor after incomplete surgery, the chance of cure with repeated resections is significantly reduced. The purpose of surgery in local recurrence is to prolong survival, although cure is probably nonrealistic since these patients present a median OS of 33 months after first recurrence [15]. Moreover, residual disease after incomplete surgery not only has a reduced DSS but also a more complex surgical approach with proven higher perioperative mortality compared to surgery for recurrent disease [16].

Reported data on morbidity and mortality after surgery for recurrent RPSs showed favorable results in terms of major morbidity (16%) and 90-day mortality (0.4%), comparable to surgery for primary disease [17]. However, there was an increased morbidity rate after vascular resections in recurrent disease that did not reach statistical significance [17]. It is important to note that morbidity may be different depending on what type of vascular resection and reconstruction is required, a matter yet to be investigated.

When planning surgery for recurrent or residual RPSs that require a major vascular reconstruction, tumor behavior together with oncological benefit, need to be balanced and discussed with the patient, addressing the surgical risk and perioperative morbidity/mortality versus predicted disease survival. The survival advantage after such an operation varies depending on tumor and treatment characteristics. To this end, recently developed RPS nomograms for recurrent disease can accurately predict the long-term disease-free and overall survival based on histology, tumor characteristics, completeness of treatment, and previous multimodal treatment [18]. In this complex MDT decision making, it is important to include the opinion of an experienced vascular or transplant surgery team, which will later be involved in the case, to have well-balanced arguments regarding risks versus oncologic benefits of an operation with such a complex vascular reconstruction [19].

## 3. Defining Tumor Vascular Involvement and Surgical Planning

### 3.1. Vascular Involvement

When defining resectability, it is important to characterize the extent of vascular resection and type of reconstruction required. The organs included in the multi-visceral resection will be determined by predicted tumor infiltration or by involvement of the primary vasculature of organs with end circulation. Major blood vessels in the retroperitoneum such as the vena cava, iliac vessels, aorta, or any major tributary can be involved at any segment in an RPS. Vascular involvement can be partial or circumferential and can be associated with or without vascular flow obstruction.

Primary vascular sarcomas of the retroperitoneum can require complex vascular reconstructions, albeit with a less extensive multi-visceral resection since they normally don't infiltrate other surrounding organs. Primary malignant vascular sarcomas of the retroperitoneum include mainly LMSs and angiosarcomas; RP-LMSs arise from the smooth muscle layer of vasculature of the retroperitoneum with a significant proportion that develop from large blood vessels such as the vena cava, iliac, or renal veins. Caval LMSs have a slight gender predominance in women and can develop at any segment of the vena cava, from the iliac vein confluence to the right atrium of the heart. Caval LMSs can be defined following Vollmann's classification depending on level of origin within the vena cava as type 1 (originating between the confluence of main iliac veins and below the level of renal veins), type 2 (originating at the level of renal veins), type 3 (originating above renal veins and below major hepatic veins), and type 4 (at the level of the major hepatic veins and right atrium) (Figure 1) [8,20].

Caval LMSs can present with symptoms secondary to vascular occlusion such as vague abdominal pain, lower extremity edema, deep venous thrombosis, or pulmonary embolism. They are biologically aggressive, with a predominant metastatic pattern of failure after curative-intent resection [1,21]. Certain studies have described a worse survival outcome of vascular originating LMS compared to other originating sites [22]. Most reports on caval LMSs are based on small patient series that do not allow for strong evidence-based conclusions; nonetheless, there is a consensus that surgery offers the best chance for prolonged survival in patients with this rare tumor [23,24].

RP-LMSs can also develop from major arterial structures such as the aorta, superior mesenteric artery, or the celiac trunk (Figure 2). Within aortic primary sarcomas, angiosarcomas represent another subset of ultrarare but aggressive tumors with poor outcomes and 5-year survival rates quoted at 8%, portraying a dismal prognosis mostly secondary to metastatic failure or tumor-related complications such as tumor embolization or ostial occlusion, resulting in mesenteric infarction [25].

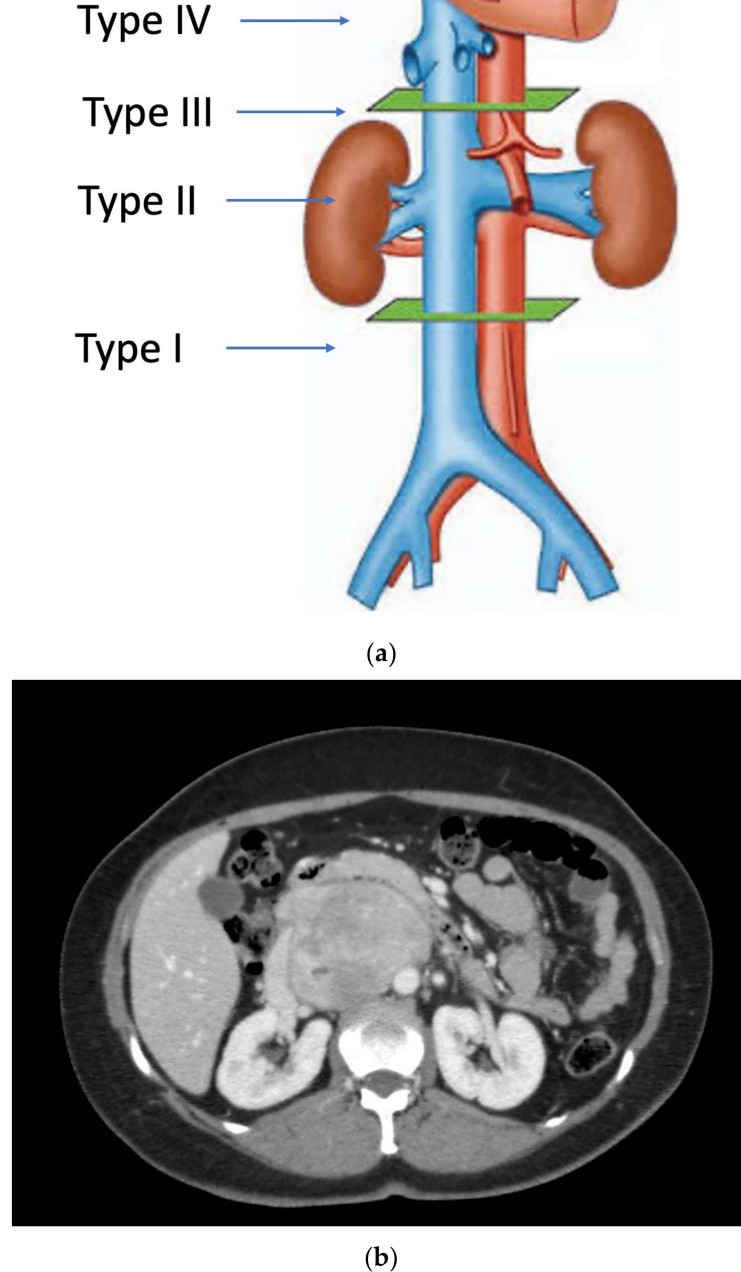

(a)

(b)

**Figure 1.** (**a**) Vollmann's IVC-LMS classification; (**b**) type 2 IVC-LMS.

*3.2. Surgical Planning*

Surgical planning for complex vascular resections is critical and mandates adequate preoperative imaging. The rational for surgical approach varies depending on the tumor histology type. RP-LPSs require a more extensive multi-visceral approach since the limit between tumor tissue and normal retroperitoneal fat cannot be accurately predicted on preoperative imaging or intraoperative exploration. This compartmental approach should include the removal of all the ipsilateral fat, colon, kidney, and psoas fascia, plus other infiltrated organs such as the pancreatic tail and major vessels. Primary RP vascular sarcomas (LMSs/angiosarcomas), on the contrary, do not require such an extensive multi-visceral approach as the tumor boundaries are better predicted on preoperative imaging.

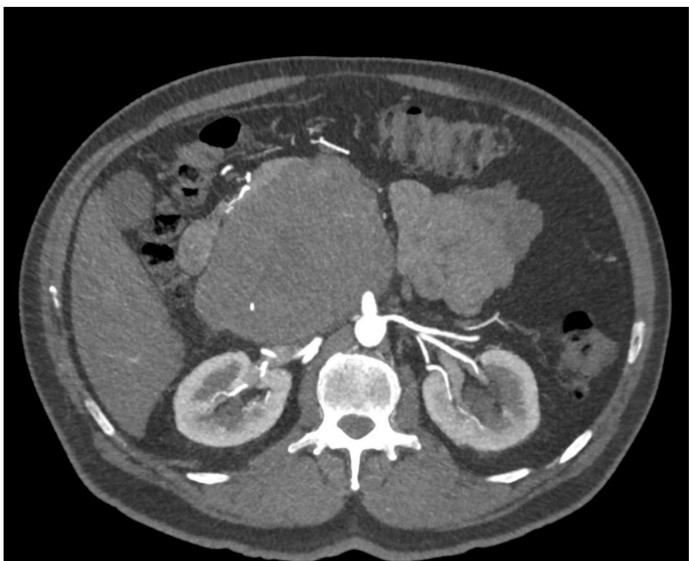

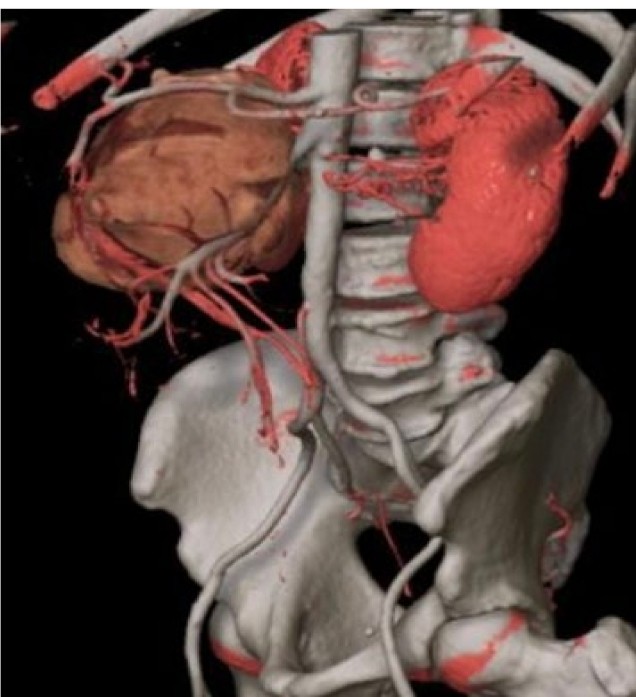

**Figure 2.** Mesenteric artery LMS: angio-CT and 3D reconstruction.

Currently, contrast-enhanced multiphase abdominal computed tomography (CT) with arterial and venous phases is preferred. Tumors arising from large vessels may be intraluminal, extraluminal, or a combination of both; therefore, contrast-enhanced cross-sectional imaging is necessary to provide a detailed evaluation of the size and extent of the tumor, as well as other characteristics such as internal tumor hemorrhage, necrosis, or cystic changes [26].

Spiral CT angiography with 3D reconstruction has become an important tool for surgical planning in tumors originating or infiltrating arterial structures such as the aorta, superior mesenteric artery, celiac trunk, or iliac arteries. The angiography can help to better predict the vascular tissue infiltration versus vascular abutment, whereas the 3D reconstruction can help to determine surrounding organ involvement and accurately plan the type of reconstruction required (Figure 2).

Abdominal magnetic resonance imaging (MRI) can be employed when a CT scan with intravenous contrast cannot be obtained. It may also help to better determine the presence of vascular collaterals when analyzing to reconstruct or simply ligate a major retroperitoneal vein following resection. The use of positron emission tomography CT (PET-CT) is not considered the standard of care for disease staging or for surgical planning in RPSs. However, several studies have explored the complementary role of PET-CT in the grading of STS [26].

## 4. Vascular Approach and Types of Reconstruction

Vascular resections can be partial or complete. The type of reconstruction, if any, will be determined by the type of vessel, extent of the tumor infiltration, and patency of vessel at the time of resection. Venous resections can be partial (side-wall resection) as they have greater compliance than arteries due to their thinner smooth muscle layer. Partial venous resections can be performed with a vascular stapler or can be performed sharply with subsequent closure of venotomy with primarily suture venorraphy or a patch using a biologic (e.g., bovine pericardium) or synthetic graft to avoid critical venous narrowing after the tumor has been resected (Figure 3).

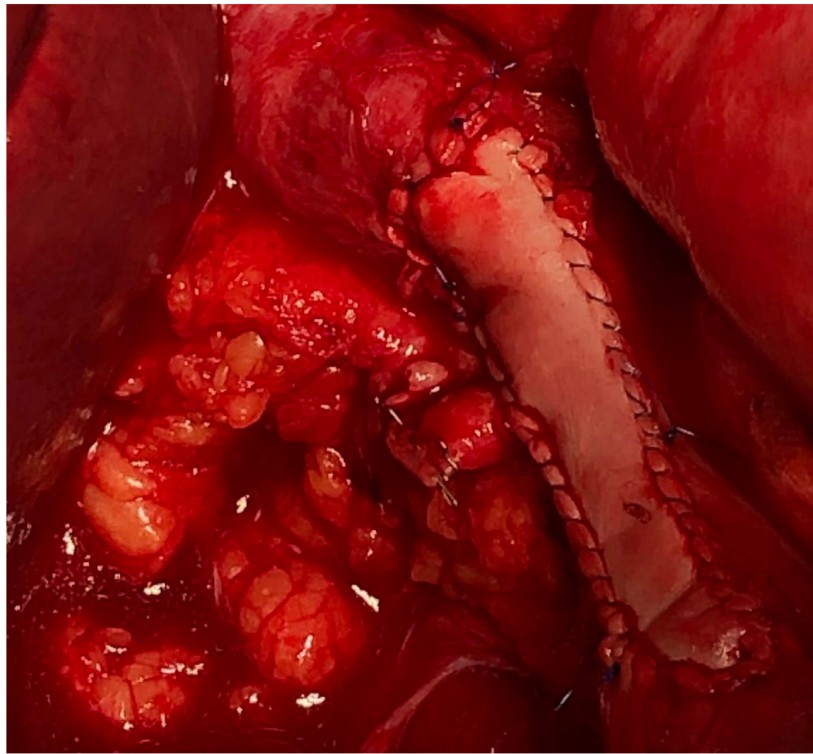

**Figure 3.** IVC partial resection with patch repair.

Arterial resection typically requires a formal segmental resection. Depending on the length of the tumor involvement, arterial repair my require an end-to-end anastomosis or an interposition bypass graft. There are many grafts options available, ranging from homologous venous grafts (e.g., saphenous, left renal, femoral, or jugular vein) to synthetic (e.g., Dacron, PTFE, and Goretex) or cadaveric grafts. The choice of graft depends on various factors: (i) clean vs. clean–contaminated case in the presence of concomitant visceral resection, (ii) availability and access to the various types of synthetic and cadaveric grafts, and (iii) surgeon preference.

*4.1. Vena Cava Reconstruction*

Reconstruction of the vena cava will vary, depending on the tumor location, extent of venous wall involvement, and the presence of chronic obstruction with established venous collateral outflow.

As aforementioned, tumor involvement of the IVC can be defined according to Vollman's classification (Types 1–4) (Figure 1).

Others have simplified this classification in only three portions: upper portion (Level 1), extending from the entry of the hepatic veins up to the right atrium; middle portion (Level 2), extending from the renal veins to the hepatic veins; lower portion (Level 3), extending from the confluence of the iliac veins to the entry of the renal veins [27]. However, the distinction between the subtype involving the hepatocaval confluence to all other retrohepatic IVC-LMSs is important, since IVC-LMSs involving segment IV are generally considered nonresectable tumors.

Suspected vena cava involvement on preoperative imaging for an RP-LPS is assessed at the time of surgical exploration. Most retroperitoneal WD-LPSs and many DD-LPSs simply displace but do not infiltrate the caval wall [28]; thus, they can be safely dissected from the vena cava adventia without need of a formal caval resection, whether partial or complete. On the contrary, caval LMSs routinely require venous resection, partial or complete; thus, a primary venorrhaphy, patch repair, or complete interposition graft may be required. Additionally, primary renal or right gonadal vein LMS can involve the caval lumen and necessitate caval repair.

4.1.1. Type of Graft

Synthetic grafts such as prosthetic polytetrafluoroethylene (PTFE) and Dacron are the most frequently used for caval replacement. They are ready accessible in centers without tissue banking, with comparable patency to biologic grafts; however, they may require life-long anticoagulation and, in the setting of a multi-visceral resection, have a higher risk of infection.

Biologic grafts include banked cadaveric vessels (e.g., IVC, iliac veins, or aorta) [8,29], autologous veins (e.g., left renal, internal jugular, and femoral), and the patient's autologous peritoneum [30]. The peritoneum graft is accomplished by dissecting a rectangular patch of the patient's peritoneum (with or without rectus abdominis posterior fascia) of a diameter that matches the caval circumference which is then sutured into a tubular shape with the mesothelial surface toward the graft's lumen [30]. Biologic grafts present the advantage of not requiring prolonged anticoagulation in the postoperative setting, as well as reduced risk of infection in the context of a gastrointestinal anastomosis as part of the multi-visceral resection. Gastrointestinal anastomotic leaks present a high risk of synthetic vascular graft infection, which is a serious complication and ultimately may require graft replacement. The patient's autologous veins are not recommended for caval replacement in contrast to other venous reconstructions such as iliac, portal, or mesenteric veins. The narrow diameter of the bypass creates a discrepancy with the IVC that can induce venous clotting.

Fiore et al. presented their results in a series of 10 patients with major caval replacement, eight of which were reconstructed using banked cryopreserved venous homografts [8]. These results proved to be a safe procedure, with low incidence of major morbidity (7%) in the hands of an experienced surgical group, in a high-volume sarcoma center. Long-term graft patency after venous homograft reconstruction was 63%, with three of eight patients presenting graft thrombosis at a median time of 7 (3–33) months post reconstruction. Fiore et al. recommended the use of venous homograft when the risk of complications was higher, in cases of extended retroperitoneal dissection or in patients who received preoperative chemoradiotherapy [8]. In terms of graft patency, the same center reported better long-term graft patency rates when reconstructing the vena cava with a PTFE graft, with a 100% 5-year patency compared to 76.7% after reconstruction with a banked venous homograft [9].

The use of a banked cryopreserved aortic graft has also been undertaken by many sarcoma centers that work together with vascular and transplant surgeons who have access to cadaveric tissue banking. An aortic homograft for caval reconstruction may present better long-term patency compared to a venous homograft, as it prevents vascular lumen to collapse due to its rigid muscular layer.

### 4.1.2. Level of Vena Cava to Reconstruct

The level at which the vena cava is resected will determine the complexity of the reconstruction when it is warranted. Resection of the vena cava below the renal veins will require only a graft interposition with cross-clamping of the IVC during the vascular anastomosis. At the level of the renal veins (i.e., juxtarenal vena cava), reconstruction of the vena cava may require the reimplantation of the renal vein to preserve kidney viability. It is important to note that resection of the cava at this level often includes en bloc resection of the right kidney. Due primarily to the anatomical length of the left renal vein and the natural venous collaterals of the left renal vein (i.e., adrenal and gonadal veins), the left kidney can be preserved when managing juxtarenal or large IVC-LMSs, as ligation of the left renal vein can be undertaken without the need for reconstruction when these collaterals can be preserved for venous outflow. However, this is not typically possible in the right kidney due to the lack of such collaterals.

Reconstruction of the vena cava above the level of the renal veins imposes another burden. This mandates mobilization of the right hemiliver including the caudate lobe to adequately expose the retrohepatic cava for vascular control and resection. Retrohepatic caval LMSs may also require an en bloc segmental liver resection.

Type 4 IVC-LMSs present the most significant technical challenge for IVC reconstruction. These tumors involve the hepatocaval confluence and, therefore, require total vascular exclusion to achieve a complete resection and reconstruction of the hepatic veins. This vascular exclusion includes portal vein and inferior vena cava inflow control and may require a veno-venous extracorporeal bypass with support of cardiac surgery team. In other cases, depending on the degree of hepatic vein involvement, resection of type 4 IVC-LMSs can require a complete liver explant with ex vivo perfusion followed by liver autotransplantation. This highly specialized procedure is performed in only a few high-volume transplant centers, and they are generally considered nonresectable due to their related morbidity and mortality.

### 4.1.3. Reconstruction Versus Ligation

Primary vascular RPSs such as caval LMSs can present with chronic venous occlusion secondary to tumoral obstruction. The complete occlusion of the vena cava is a slow but progressive process of caval LMSs, which in turn leads to the development of collateral venous outflow via the gonadal, lumbar, retroperitoneal, and abdominal wall veins. Surgical planning and preoperative imaging for caval LMSs with complete lumen occlusion should specifically investigate for venous collaterals, as this may preclude the need to reconstruct the vena cava and only ligate it after en bloc resection.

Ligation of the vena cava (at any level) in patients with good collateral venous outflow presents some advantages compared to reconstruction. There is no need for life-long anticoagulation compared to synthetic grafts due to the reduced risk of pulmonary embolism as there is no chance of caval graft thrombosis [29,31]. However, there are side-effects after caval ligation such as pelvic girdle and lower-extremity edema, symptomatic lower-extremity deep venous thrombosis, and acute kidney injury; however, these are typically acute postoperative complications that resolve with time. Certain authors argue that the vena cava should not be reconstructed at all after resection for RPSs due to the variable vascular patency after graft reconstruction [31].

### 4.2. Iliac Vessels

Iliac vessels, including the vein and artery, can be compromised either by retroperitoneal sarcomas projecting into the pelvis or by pelvic sarcomas. The most common histologic subtypes are RP-LMSs and RP-LPSs, as well as others more frequently found in the pelvis including solitary fibrous tumors, synovial sarcoma, MPNST, or Ewing's sarcoma [8,9]. Iliac vessels (vein and artery) are anatomically in closer proximity; therefore, it is not uncommon that both vessels may be involved in the tumor mass, requiring a combined artery and vein reconstruction.

The iliac artery reconstruction is fashioned similarly with synthetic or biologic graft material. The internal iliac artery can be ligated and not reimplanted to the graft. If the graft crosses the inguinal ligament to be anastomosed to the femoral artery, consideration of a musculocutaneous or myofascial flap to cover and protect the synthetic graft from superficial exposure must be undertaken [32]. When iliac vessels are reconstructed with a concomitant visceral resection, consideration for an extra-anatomical bypass (femoral artery to femoral artery) can be performed if there is an issue with availability and access to biologic grafts or the surgeon believes the risk of graft infection is too high to consider using a synthetic graft. Another possibility in the absence of banked tissue is to harvest the contralateral femoral artery or vein, reconstructing it with synthetic graft, and using the autologous graft to replace the resected iliac artery.

Management of the iliac vein follows similar dogmas as caval resections: patch repair, interposition graft, or ligation. The iliac vein is divided into two segments to plan reconstruction: segment I is defined between the origin of the external iliac vein and the internal iliac/hypogastric vein to the ilioinguinal ligament; segment II is defined by the common iliac vein per se. Reconstruction of the common or external iliac veins can be fashioned with an autologous left renal vein, contralateral femoral vein, cryopreserved cadaveric graft, or PTFE graft. Patency after reconstruction is lower compared to IVC reconstruction, favoring the use of PTFE with a 71% long-term patency [9] (Figure 4). When iliac vein resection includes the internal iliac inflow or hypogastric vein, this vein can simply be ligated.

### 4.3. Aortic Replacement

Primary sarcomas of the aorta can be categorized by histology and according to where they appear to arise (i.e., aortic media or intima), among which intimal sarcoma types are the most common [33]. In the context of RPSs with major vascular involvement, the aorta is generally infiltrated by high-grade nonvascular sarcomas such as DD-LPSs.

Oncovascular replacement of the abdominal aorta is generally performed at the infrarenal aorta level. The circumferential tumor involvement of the suprarenal aorta implies a very high-risk procedure due to the debranching and reimplantation of renal and mesenteric arteries (e.g., superior mesenteric artery and celiac trunk). Considering that this aortic reconstruction could include a multi-visceral reconstruction for primary/recurrent high-grade nonvascular RPSs (DD-LPSs), the perioperative mortality most certainly would surpass the oncologic survival benefit; therefore, circumferential suprarenal aortic tumor involvement for nonvascular primary disease is generally considered nonresectable.

In primary aortic STSs, such as angiosarcomas, resection and reconstruction of the aorta may not require a multi-visceral resection. Therefore, the risk of suprarenal aortic reconstruction may be undertaken considering it is the only therapeutic option for a primary aggressive tumor.

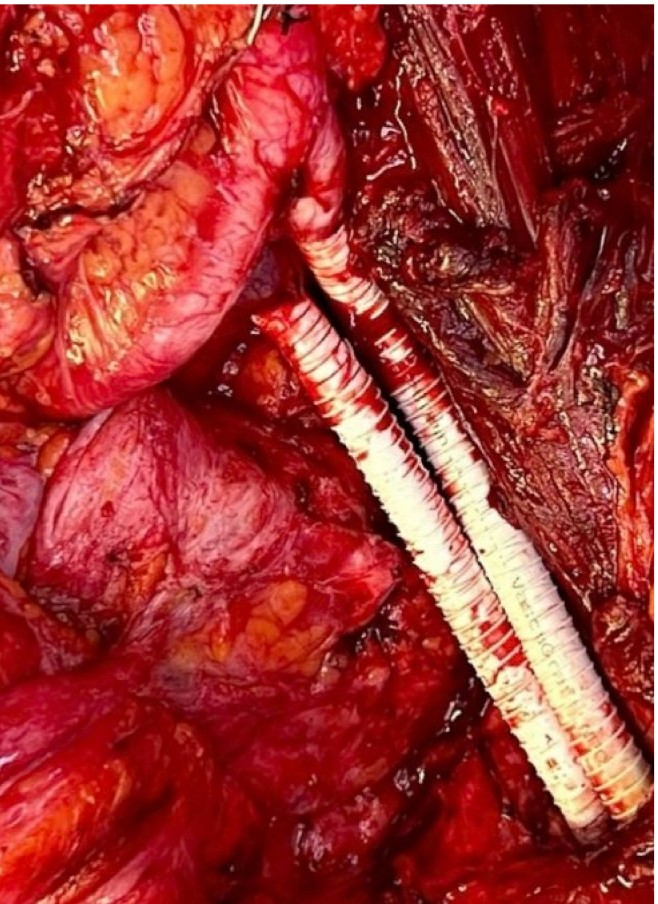

**Figure 4.** Iliac artery and vein reconstruction after DD-LPS resection.

## 5. Postoperative Complications after Major Vascular Reconstruction

Complications in RPS surgery may vary, and the association with major vascular reconstruction is correlated with a significant increase in postoperative morbidity (odds ratio = 2.14) [34]. The experiences reported from different high-volume centers show a morbidity rate ranging from 7% to 36% and a postoperative mortality ranging from 0% to 21% (Table 1).

**Table 1.** Clinicopathological characteristics and perioperative results of case series on major vascular reconstructions for RPS.

| Study | Patients (N) | Years | Histologic Subtypes of RPS | Tumor Grade | Primary/Recurrent | Neoadjuvant Treatment | Type of RP Vessel Involved | Type of Reconstruction | Type of Graft Repair | Follow up (Months) | Patency Rate | Morbidity | Mortality |
|---|---|---|---|---|---|---|---|---|---|---|---|---|---|
| Ridwelski et al. [20] | 5 | 1993–1999 | LMS | NS | 5/0 | NS | IVC = 5 | Graft = 3 | PTFE = 3 | NS | NS | NS | NS |
| Hollenbeck et al. [24] | 25 | 1982–1992 | LMS | G3 = 25 | 25/0 | Chemo = 4 | IVC = 25 | Ligation/repair = 19 Graft = 2 | PTFE = 2 | 24 | 80% | 4/25 (16%) | 2/25 (8%) |
| Schwarzbach et al. [32] | 25 | 1988–2004 | LMS = 12 LPS = 4 MFH = 2 CCS = 2 NS = 5 | G1 = 2 G2 = 5 G3 = 17 NS = 1 | 18/7 | Chemo = 3 | AO = 6 IA = 3 IVC = 11 SMV = 2 IV = 3 | Ligation/repair = 7 Re-anastomosis = 1 Graft = 17 | PTFE = 9 Dacron = 8 | 19.3 | 89% | 9/25 (36%) | 1/25 (4%) |
| Kieffer et al. [35] | 19 | 1979–2004 | LMS = 19 | NS | 19/0 | Chemo = 6 Radiation = 1 | IVC = 19 | Ligation/repair = 6 Graft = 13 | PTFE = 13 | 43.2 | 90% | NS | 4/19 (21%) |
| Ito et al. [36] | 20 | 1990–2006 | LMS = 20 | G1 = 2 G2 = 7 G3 = 10 NS = 1 | 20/0 | Chemo = 2 Radiation = 6 Chemo + Rad = 1 | IVC = 20 | Ligation/repair = 15 Graft = 5 | Synthetic = 5 (NS) | 40.8 | NS | NS | 0/20 |
| Fiore et al. [8] | 15 | 2004–2011 | LMS = 12 DD-LPS = 1 SFT = 1 UPS = 1 | NS | 15/0 | Chemo = 9 Radiation = 6 | IVC = 15 | Ligation/repair = 5 Graft = 10 | PTFE = 2 Banked homograft = 8 | 31.6 | 60% | 1/15 (7%) | 0/15 |
| Cananzi et al. [23] | 11 | 2000–2012 | LMS = 11 | G1 = 3 G2 = 3 G3 = 4 NS = 1 | 11/0 | Chemo = 4 Chemo + Rad = 2 | IVC = 11 | Ligation/repair = 7 Graft = 4 | Synthetic = 4 NS | 60 | NS | 4/11 (36%) | 0/11 |
| Ferraris et al. [9] | 67 | 2000–2016 | LMS = 42 DD-LPS = 13 WD-LPS = 4 MPNST = 3 SFT = 1 Others = 2 | G1 = 7 G2 = 29 G3 = 31 | 62/5 | NS | IVC = 39 IV = 24 IVC + IV = 4 AO + IA = 17 | Ligation/repair = 30 Graft = 58 | PTFE = 33 Banked homograft = 24 Autologous graft = 1 | 57.6 | PTFE = 100% Homograft = 76% | 15/67 (22.4%) | 2/67 (3%) |
| Homsy et al. [19] | 17 | 2010–2018 | LMS = 9 DD-LPS = 3 Myxoid LPS = 1 Sclerosing LPS = 1 UPS = 1 AS = 1 Other = 1 | G2 = 6 G3 = 10 | 11/6 | NS | AO = 7 HA = 1 CT = 1 SMA = 1 IA = 3 IVC = 9 IV = 5 | Ligation/repair = 1 Graft = 16 | PTFE = 9 Banked homograft = 5 Autologous graft = 9 | | 64% | 5/17 (29%) | 0/17 |

LMS = leiomyosarcoma, LPS = liposarcoma, DD-LPS = dedifferentiated liposarcoma, WD-LPS = well-differentiated liposarcoma, SFT = solitary fibrous tumor, UPS = undifferentiated pleomorphic sarcoma, MFH = malignant fibrous histiocytoma, CCS = clear-cell sarcoma, MPNST = malignant peripheral nerve sheath tumor, AS = angiosarcoma, NS = not specified, AO = aorta, SMA = superior mesenteric artery, HA = hepatic artery, CT = celiac trunk, IA = iliac artery, IVC = inferior vena cava, SMV = superior mesenteric vein, IV = iliac vein.

Many technical factors may predict the possibility of a complication after major vascular reconstruction in the setting of RPS surgery. These include the extent of multi-visceral resection (number of organs resected) [17,34], the use of neoadjuvant radiation, the type of vessel to be reconstructed, the type of reconstruction required, and the type of graft used [8].

Complications can also vary in the short- and long-term postoperative follow-up. Most common complications in the immediate postoperative period are hematomas, surgical site bleeding, and renal failure [9]. These complications are generally related to the need of anticoagulation for graft patency in the immediate postoperative recovery and renal outflow clamping in the setting of renal vein or IVC reconstruction.

In the intermediate and long-term follow up, the most common complication is vascular graft thrombosis. Reported graft patency after reconstruction also varies from 60–100% (Table 1), with recognized factors such as type of vessel and graft used. Smaller vein reconstructions, e.g., for iliac veins, have a higher thrombosis rate when synthetic grafts are used. Therefore, the interposition of an autologous vein is generally preferred.

Another serious complication, as previously mentioned, is graft infection. This complication is generally associated with a multi-visceral resection where the graft stays in proximity with visceral anastomosis. The use of neoadjuvant radiation may increase the chances of small or large bowel anastomotic leak; hence, the use of biologic grafts (autologous or banked homograft) of extracorporeal bypass may significantly reduce this complication.

## 6. Areas of Doom—A Possible Role for Autotransplantation in RPSs?

Perioperative morbidity and oncologic results after complex surgery for rare tumors significantly improve when patients are treated in specialized high-volume centers. This has been particularly evident in retroperitoneal sarcomas [37–39]. The complexity of each tumor subtype and the role of personalized multimodal treatment have only been understood thanks to collaborative research [1,34] led by the experience in centers that focus on the care of rare diseases.

Considering that surgery is still the mainstay for cure, major efforts have been set to increase the chances of resectability in primary RPSs even when major vascular resections are involved.

From a purely surgical aspect, the experience shared among high-volume centers has led to an understanding of the extent of resection required to secure adequate oncologic margins, with a current perspective toward a more tailored resection extent [28].

There are certain vascular reconstructions, however, that are significantly morbid even in reference centers, with high perioperative morbidity and mortality, i.e., the replacement of the superior mesenteric artery (SMA) and the reconstruction of major hepatic veins at the hepatocaval confluence.

Within aggressive epithelial carcinomas (pancreas cancer and primary/metastatic liver tumors), tumors involving the SMA or the hepatocaval confluence have historically been considered nonresectable. This is due to the high morbidity and perioperative mortality associated with the resection, which defeats any oncologic benefit. The complexity of this vascular reconstruction relies on the difficult anatomic exposure of an area of vessels that carry high blood flow, to and from the heart, and where any major blood loss rapidly determines critical pressure instability, prolonged warm ischemia, and secondary organ ischemic damage to the small bowel or the liver.

The development of new chemotherapy drugs and the increasing experience in oncovascular surgery have raised hopes of improving resectability of tumors that involve these structures, particularly in slow-growing tumors with prolonged overall survival such as some RPS histologic subtypes. The better selection of patients with promising survival after good response to chemotherapy and patients with indolent histologies considered unresectable for SMA involvement has prompted physicians to undertake risks associated with this procedure while developing an experience in SMA reconstruction for oncologic surgery [19,40]. Published experiences in this procedure for RPSs are mostly based on

single case reports or small patient series. The strongest experience comes from surgery for pancreas cancer, reporting major perioperative morbidity of 28%, and 90-day postoperative mortality of 15% [41,42].

Reconstruction of the hepatocaval confluence for liver tumors has also been reported with similar rates of perioperative morbidity and mortality, most of which include patients with aggressive liver primary or metastatic carcinomas. In a series of 37 patients treated for primary and metastatic liver tumors with IVC infiltration, Li et al. reported a total of 17 patients who required resection and reconstruction of the retrohepatic IVC at the hepatocaval confluence. Perioperative morbidity of this series was 40.7% with a perioperative mortality of 16.7%, including a higher mortality among those 17 patients with hepatic vein reconstruction.

The possibility of organ autotransplantation could potentially reduce the time of warm ischemia after complete vascular exclusion during these complex oncovascular reconstructions. Autotransplantation for oncologic resections has been reported as a way of maximizing vascular control with improved exposure for arterial reconstruction with total vascular exclusion [43,44]. The complete organ exenteration with in situ cold perfusion using preservation solutions enables a controlled tumor resection during back table surgery, with reduced blood loss and minimal organ warm ischemic damage. Tzakis et al. reported a series of 10 patients with intestinal and multi-visceral autotransplantation for tumors of the root of the mesentery, including desmoid tumors and RP-LMSs, with long-term survival ranging from 13–138 months [45]. The physiologic foundation of partial exenteration, ex vivo resection, and intestinal autotransplantation is the use of cold preservation solution, which allows complete tumor resection followed by a complex vascular reconstruction in a bloodless field with minimum ischemic damage to the explanted organ.

In an evolving era of transplant oncology, organ autotransplantation may potentially increase resectability for select tumors arising in anatomic areas that are currently considered nonresectable. This requires further investigation and the combined work of surgical oncologists with highly trained multi-visceral transplant teams, which is not a reality in many high-volume sarcoma centers.

## 7. Conclusions

The improved knowledge in tumor biology, including a better understanding of the local behavior, patterns of failure, and survival in RPSs, enables today a multimodal tailored approach for primary and recurrent disease. The histology-based approach helps to define the required extent of the surgical resection, including major vascular resection, that can provide the best survival benefit for the patient. The concentration of surgical volume in specialized sarcoma centers has enabled securing adequate results in terms of morbidity and mortality after highly complex vascular resections and reconstructions.

**Author Contributions:** Conception and design, N.A.D. and K.C.; Administrative support, K.C.; Provision of study materials or patients, N.A.D.; Collection and assembly of data, N.A.D.; Data analysis and interpretation, N.A.D., J.M.B., J.C.D.l.M., S.v.M.H. and K.C.; Manuscript writing, N.A.D., J.M.B., J.C.D.l.M., S.v.M.H. and K.C.; Final approval of manuscript, N.A.D., J.M.B., J.C.D.l.M., S.v.M.H. and K.C. All authors have read and agreed to the published version of the manuscript.

**Funding:** This research received no external funding.

**Institutional Review Board Statement:** The study did not require ethical approval.

**Informed Consent Statement:** No informed consent was required for this review.

**Conflicts of Interest:** The authors declare no conflict of interest.

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
