# Peer review of "Complex Vascular Resections for Retroperitoneal Soft Tissue Sarcoma"

_curroncol, doi:10.3390/curroncol30030266_

Round 1
Reviewer 1 Report
The topic is very interesting.
However, this narrative review is disorganized.
I suggest to add a review of the Literature (with a relative table) on studies reporting on this topic. Do the Authors have their own series to report?
Also, please report on complication. In particular, what about post radiotherapy reconstructions? May silver coated prosthesis be used?
In addition, please discuss on multidisciplinary approach to these patients (eg Sambri et al, Curr Inc 2023). The role of vascular surgeon should be integrated on a multidisciplinary team.
Author Response
Dear reviewer:
Thank you very much for your thoughtful comments with thorough suggestions to improve this Review.
In response to these, I have included a Table summarizing the most significant experience reported in vascular reconstruction for RPS, excluding those that include extremity vascular repair. We do have our institutional data collected with 10 cases of major vascular reconstruction over a period of 4 years. I could include this data in the table as a "not published experience" if you think it adds to the review coming from a center in South America, however I have only included at this point published data.
I have included further detail about Vascular/Transplant Surgery teams in the specialized Sarcoma MDT discussion.
Finally, regarding your suggestion of giving more details in perioperative complications, I have included a section in the article to discuss this matter in further detail, including factors involved to define this complications.
Please find attached my final version.
Kind regards
Nicolas Devaud

Reviewer 2 Report
The authors present a very detailed, comprehensive review of vascular reconstructions in RPS. It is a very well written manuscript with an appropriate emphasis on the biology of these tumors, which is routinely weighted heavily in any decisions regarding vascular resection/reconstruction. The breadth and detail of the material are excellent and the technical considerations presented are appreciated. Overall, it is a very nice review and will be of great value to the community of physicians treating sarcoma.
One minor comment is with respect to the discussion of managing the iliac vein. The authors describe the system utilized to anatomically divide this region as it relates to resection/reconstruction, but then solely identify situations where the iliac vein can be ligated with impunity. Can the authors expand the discussion of when they consider reconstructing the iliac vessels, and if so what are the technicalities which weigh into that decision?
Lastly, there are minor grammatical errors throughout the manuscript. It should be carefully proofread prior to final submission. I provide a few examples below:
Line 183 – on the hand is likely meant to say “on the other hand”
Line 199-200 – grammar error “…as well as be safe to use…”
Line 208 – grammar error “These results proved to be a safe procedure…”
Line 322-323 – grammar error “From a purely surgical aspect, the experience shared among high-volume centers has led to understand better the extent of the resection that’s required…”
Line 357-358 – grammar error “during this complex oncovascular reconstructions...”
Author Response
Dear reviewer:
Thank you very much for your kind comments and thoughtful suggestions to improve this review. We have edited all grammar errors you have accurately identified. We have also clarified the approach to reconstruct iliac veins and when to consider just ligation.
Please find attached our edited version of the manuscript.
Kind regards,
Nicolas Devaud

Reviewer 3 Report
Thank you very much for allowing me to review this paper. It is a very interesting topic on sarcoma surgery, and there are very few previous reviews addressing this subject in such a deep and comprehensive way. So, I must congratulate the authors for this work. I really like the way the authors have articulated the review, favoring its usefulness for oncological and vascular surgeons interested on sarcoma surgery.
I have some minor comments of the following questions:
Comment 1
- Introduction. Row 35: referring to sarcomas of infiltrative pattern I would add to DD-LPS other histologies where a vascular resection may be needed, such as UPS, SFT or MPNST. Is true that these histologies are referred further in the text, but I think it is important to make clear in the introduction that vascular resection is not exclusively performed in DD-LPS and-or LMS.
Comment 2
- Row 63. The author must be more specific describing the role of neoadjuvant treatment. They say that “certain cases” may benefit from preoperative treatment… they should make clear that it is usually employed on borderline tumors in the limit of resectability to favor R0/R1 resections or in tumors with a high risk of distant metastases to select those patients who may benefit from such an aggressive surgery. They should also make a comment on the role of preoperative radiotherapy (RT) (STRASS 1 and its secondary analyses), as some master authors (Bonvalot, Gronchi) has stated that preop.RT may increase the rate of R0 resections preventing vascular resections because it favors a periadventicial dissection by reducing marginality in these critical structures.
Comment 3
- Section 3. Row 107. The authors should clear out the main differences on the rational for the surgical approach between LPS and LMS/angiosarcoma. LPS surgery should aim to remove all the ipsilateral fat and entails a liberal resection of the colon, kidney and psoas fascia, plus other infiltrated organs (pancreatic tail, abdominal vessels etc). On the contrary for primary vascular sarcomas we should perform an en bloc resection of the tumor with a segmentary resection of the vein/artery, and just resect the infiltrated organs to avoid R2 resections.
-
Comment 4
- Row 115. The authors must mention that they are referring to Vollmann’s classificaction of ICV LMS and they should describe if there are other classification systems (as Kulaylat’s: Kulaylat MN, Karakousis CP, Doerr RJ, Karamanoukian HL, O'Brien J, Peer R. Leiomyosarcoma of the inferior vena cava: a clinicopathologic review and report of three cases. J Surg Oncol. 1997;65:205-17)
- Fig 1. I would recommend to change the text for fig a. ïƒ Vollmanns’ classification of IVC-LMS, and the diagram should be referenced
Comment 5
- Section 3.2 Surgical Planning: I consider that a deepest review on the role of preoperative imaging assessing for vascular involvement is missed. The authors should be more exhaustive on the role of preoperative imaging and the critical aspects that our Radiologist should report when evaluation a big retroperitoneal tumor with possible vascular involvement. Is there a role for CT angiography? And for 3D reconstruction?
Comments 6
- Section 4. Row 168 there is a typo in the word Goretex. e.g., Dacron, PTFE, Gortex, etc.)
- Section 4.1.2 Level of vena cava reconstruction. Regarding type 4 IVC LMS the authors should explain that such resections usually demand a vascular exclusion, with or without veno-venous bypass, and sometimes in cooperation with a cardiac surgeon.
Comment 7
- Section 4.2 Iliac vessels. The rational for external iliac vein ligation vs reconstruction is not clear. Please, consider to re-write this part (row 281-293)
Author Response
Dear reviewer:
Thank you very much for your thoughtful comments and thorough suggestions to improve our Review. We have followed them all in detail (comments 1-7 and you may find them accurately edited in our latest version of the manuscript that we have attached.
Again, thank you very much for taking the time for such a detailed analysis of our Review.
Kind regards,
Nicolas Devaud

Round 2
Reviewer 1 Report
Thank you to the Authors for the attempt to ameliorate their paper.
It now merits publication.
I would add two references for a comparison with vascular reconstruction in other sites STS.
Angelini et al. ( doi: 10.3390/jpm11060462)
Sambri et al. (doi: 10.3390/curroncol30010084)